# CHART AUTO-ENCODERS FOR MANIFOLD STRUCTURED DATA

## ABSTRACT

Auto-encoding and generative models have made tremendous successes in image and signal representation learning and generation. These models, however, generally employ the full Euclidean space or a bounded subset (such as $[0, 1]^l$) as the latent space, whose flat geometry is often too simplistic to meaningfully reflect the topological structure of the data. This paper aims at exploring a universal geometric structure of the latent space for better data representation. Inspired by differential geometry, we propose a **Chart Auto-Encoder (CAE)**, which captures the manifold structure of the data with multiple charts and transition functions among them. CAE translates the mathematical definition of manifold through parameterizing the entire data set as a collection of overlapping charts, creating local latent representations. These representations are an enhancement of the single-charted latent space commonly employed in auto-encoding models, as they reflect the intrinsic structure of the manifold. Therefore, CAE achieves a more accurate approximation of data and generates realistic synthetic examples. We demonstrate the efficacy of CAEs through a series experiments with synthetic and real-life data which illustrate that CAEs can out-preform variational auto-encoders on reconstruction tasks while using much smaller latent spaces.

## 1 INTRODUCTION

Autoencoding (Bourlard & Kamp, 1988; Hinton & Zemel, 1994; Liou et al., 2014) is a central tool in unsupervised representation learning. The latent space therein captures the essential information of a given data set, serving the purposes of dimension reduction, denoising, and generative modeling. Even for models such as generative adversarial networks (Goodfellow et al., 2014) that do not employ an encoder, the generative component starts with a latent space. A common practice is to model the latent space as a low-dimensional Euclidean space $\mathbb{R}^l$ or a bounded subset of it (e.g., $[0, 1]^l$), sometimes equipped with a prior probability distribution. Such spaces carry far simple geometry and may not be adequate for representing complexly structured data. In this work, we are concerned with a widely studied structure: manifold.

A commonly held belief, known as the *Manifold Hypothesis* (Belkin & Niyogi, 2003; Fefferman et al., 2016), states that real-world data often lies on, or at least near, some low-dimensional manifold embedded in the high-dimensional ambient space. Hence, a natural approach to representation learning is to introduce a low-dimensional latent space to which the data is mapped. It is desirable that such a mapping possesses basic properties such as invertibility and continuity. In differential geometry, such a notion is coined *homeomorphism*. Challengingly, it is known that even for some simple manifolds, there does not always exist a homeomorphic mapping to the Euclidean space of the intrinsic dimension of the data. We elaborate two examples such examples next.

Consider a data set $X$ lying on the 2-dimensional sphere $S^2$ embedded in the ambient space $\mathbb{R}^n$ where $n \geq 3$. It is well known that there exist no homeomorphic maps between $S^2$ and an open domain on $\mathbb{R}^2$ (Rotman, 2013). Therefore, it is impossible for a traditional autoencoder with a 2-dimensional latent space to faithfully capture the structure of the data. Consequently, the dimension of the latent space needs be increased beyond the intrinsic dimension (two in this case). For another example, we show in Figure 1 a double torus. When one uses an autoencoder to map uniform data points on this manifold to $\mathbb{R}^2$, the distribution of the points is distorted and the shape destroyed, whereas if one maps to $\mathbb{R}^3$, some of the points depart from the mass and become outliers. Fur-

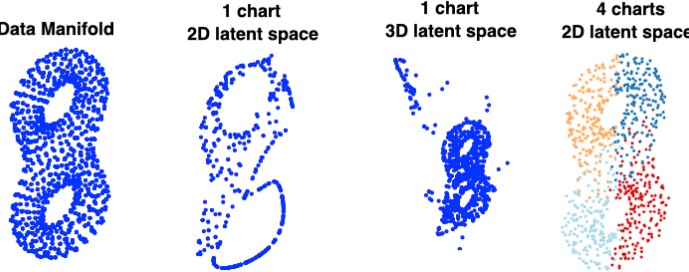

Figure 1: Left: Data on a double torus. Middle two: Data autoencoded to a flat latent space. Right: Data autoencoded to a 4-chart latent space.

thermore, in the appendix (see Figure 11) we demonstrate that increasing the number of parameters of the autoencoder does not help overcome the coverage issue when the latent space is a single 2-dimensional space.

To better reflect structures, in this work, we follow the definition of manifolds in differential geometry and propose **Chart Auto-Encoder (CAE)** for learning a low-dimensional representation of data lying on a manifold. Rather than using a single function mapping, the manifold is parameterized by a collection of overlapping *chart functions*, each of which describes a local neighborhood and they collectively cover the entire manifold. To the right of Figure 1, we show the same double torus aforementioned, now encoded by using four color-coded charts. One sees that the encoding result faithfully preserves the shape of the data set, as well as the topology (two holes).

To realize the parameterization, we develop a neural network architecture and propose a training regime to implement it. We conduct a comprehensive set of experiments on both synthetic data and real-world data to demonstrate that CAE captures much better the structure of the data and enriches the understanding of them.

## 1.1 RELATED WORKS

One common approach to enhancing the capability of autoencoders is to impose a prior distribution on the flat Euclidean latent space, as in variational autoencoders (VAE) (Kingma & Welling, 2013). The distributional assumption (e.g., Gaussian) introduces low-density regions that sometimes depart from the manifold. Then, paths in these regions will either trace off the manifold or become invariant. Falorsi et al. (2018) introduce a non-Euclidean latent space to guarantee the existence of a homeomorphic representation by using a so-called homeomorphic variational autoencoder (HVAC). There are two disadvantages of this approach. First, it requires the user to know the topological class of the data set, which can be extremely difficult for real-world data. Secondly, it requires the computation (or estimation) of the Lie group action on the latent space. If the topology of the data is relatively simple (e.g., a sphere or torus), the computation is possible, but for more complex objects the estimation is much difficult. Similarly, several recent works (Davidson et al., 2018; Rey, 2019; Falorsi et al., 2018) have studied autoencoders with (hyper)-spherical latent spaces. These methods allow for the detection of cyclical features, but offer little insight into the homology of the manifold, since it will always be represented by a compact genus-zero surface.

Exploring the low-dimensional structure of manifolds has led to many dimension reduction techniques in the past two decades (Tenenbaum et al., 2000; Roweis & Saul, 2000; Cox & Cox, 2001; Belkin & Niyogi, 2003; He & Niyogi, 2003; Zhang & Zha, 2004; Kokiopoulou & Saad, 2007; van der Maaten & Hinton, 2008). Isomap (Tenenbaum et al., 2000) divides a data set into local neighborhoods, which are embedded into a low-dimensional space that preserves local properties. Similarly, Laplacian Eigenmaps (Belkin & Niyogi, 2003) use embeddings induced by the Laplace-Beltrami eigen-functions to represent the data. These methods employ a flat Euclidean space for embedding and may lose information as aforementioned. Recently, Lei et al. (2019) considers a manifold point of view to explain Wasserstein generative adversarial network (WGAN) using optimal transport by minimizing a distance between a manifold parameterize by a neural network and one estimated from some training data.

Under the manifold hypothesis Chen et al. (2019) extend the work of Shaham et al. (2018) and show the theoretical existence of neural networks that approximate functions supported on low-dimensional manifolds, with a number of parameters only weakly dependent on the embedding dimension. A key feature in their proposal is a chart determination sub-network that divides the manifold into charts and a pairing sub-network that re-combines them. All existing methods only have theoretical consideration of chart structure for data manifolds and assume that the manifold in question is known. However, multi-chart latent space representations have not been implemented and conducted computationally. Moreover, part of the constructions relies on information of the underlying manifold, which is unavailable in most real-world applications. Additionally, important questions regarding the loss function and the training method remain open. Our work introduces an implementable neural network architecture which is able to address these challenges by directly approximating the data manifold.

## 2 BACKGROUND ON MANIFOLDS AND CHART BASED PARAMETERIZATION

A manifold is a topological space locally homeomorphic to a Euclidean domain. More formally, a $d$-dimensional manifold is defined as a collection of pairs $\{(\mathcal{M}_\alpha, \phi_\alpha)\}_\alpha$, referred to as *charts*, where $\{\mathcal{M}_\alpha\}_\alpha$ are open sets satisfying $\mathcal{M} = \bigcup_\alpha \mathcal{M}_\alpha$. Each $\mathcal{M}_\alpha$ is homeomorphic to an open set $U_\alpha \subset \mathbb{R}^d$ through a coordinate map $\phi_\alpha : \mathcal{M}_\alpha \to U_\alpha$. Different charts may be glued together through transition functions $\phi_{\alpha\beta} : \phi_\alpha(\mathcal{M}_\alpha \cap \mathcal{M}_\beta) \to \phi_\beta(\mathcal{M}_\alpha \cap \mathcal{M}_\beta)$ satisfying cyclic conditions (see Figure 2 left). Smoothness of the transition functions controls the smoothness of the manifold. A well-known result from differential geometry states that any compact manifold can be covered by a finite number of charts which obey these transition conditions. The *intrinsic dimension* of the manifold is the dimension of $U_\alpha$. See Lee (2013) for a thorough review.

In practice, the coherent structure of data motivates us to model a given data as samples from an unknown ground manifold. One crucial task in machine learning is to explore the topological and geometric structure of the manifold and perform tasks such as classification and data generation. Mathematically, we explain the encoding and decoding process for a manifold as follows. Given some manifold $\mathcal{M}$, usually embedded in a high dimension ambient space $\mathbb{R}^n$, the encoding network constructs a local parameterization $\phi_\alpha$ from the data manifold to the latent space $U_\alpha$; and the decoding network maps $U_\alpha$ back to the data manifold $\mathcal{M}$ through $\phi_\alpha^{-1}$. In standard autoencoders, only one single chart $U_\alpha$ is used as the latent space. In our work, multiple charts are used. Different from classical dimension reduction methods where distance preservation is preferred, we do not require the local parameterization $\phi_\alpha$ to preserve metric but only bound its Lipschitz constant to control the regularity of the parameterization (see Section 4 for more details).

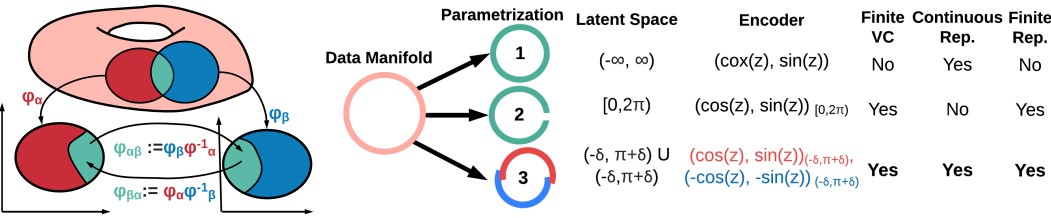

Figure 2: Left: Illustration of a manifold. Right: Possible parameterizations of a circle. The manifold approach (bottom) preserves all desired properties.

To illustrate the utility of such a multi-charted parameterization, we consider a simple example: find a latent representation of data sampled from a 1-dimensional circle $S^1$ embedded in $\mathbb{R}^2$. See Figure 2 right. A simple (non-chart) parameterization is $(\cos(z), \sin(z))$ with $z \in (-\infty, \infty)$. However, approximating this parameterization with a finite neural network is impossible, since $z$ is unbounded and hence any multi-layer perceptron will have an infinite Vapnik-Chervonenkis dimension (Blumer et al., 1989). One obvious alternative is to limit $z \in [0, 2\pi)$, but this parameterization introduces a discontinuity and breaks the topology (it is theoretically known that the closed circle is not homeo-

morhpic to $[0, 2\pi)$). Following the definition of manifold, we instead parameterize the circle as:

$$
\begin{aligned}
\phi_\alpha &: (0 - \delta, \pi + \delta) \to S^1, & z_\alpha &\mapsto (\cos(z), \sin(z)) \\
\phi_\beta &: (0 - \delta, \pi + \delta) \to S^1, & z_\beta &\mapsto (-\cos(z), -\sin(z)) \\
\phi_{\alpha\beta} &: (-\delta, \delta) \to (\pi - \delta, \pi + \delta) & z_\alpha &\mapsto z_\alpha + \pi \\
\phi_{\alpha\beta} &: (\pi - \delta, \pi + \delta) \to (-\delta, \delta) & z_\alpha &\mapsto z_\alpha - \pi
\end{aligned}
\tag{1}
$$

Although this function is more cumbersome to write, it is more suitable for representation learning, since each encoding function can be represented with finite neural networks. Moreover, the topological and geometric information of the data is maintained.

Thus, instead of using only one chart as in standard autoencoders (Bourlard & Kamp, 1988; Hinton & Zemel, 1994; Liou et al., 2014), we propose to model the latent space with multiple charts glued by their transition functions, akin to the concept of manifold. This geometric construction reflects the intrinsic structure of the manifold. Therefore, it is able to achieve more accurate approximation of the data and generate realistic new ones. Moreover, once the charts and the associated transition functions are learned, the geometric information of the data manifold, including metric, geodesics, and curvatures, can be approximated according to their definitions in differential geometry. Thus, this multi-chart latent construction leads to a better geometric understanding of the manifold.

## 3 NETWORK ARCHITECTURE

To integrate manifold structure in the latent space, we investigate a new representation of latent space based on a multi-chart structure. We implement a novel network architecture to learn the multi-chart latent space and its transition functions. The proposed network architecture, illustrated in Figure 3, can be summarized as follows: An input data $x$ is passed into an encoding module, which creates an initial latent representation. Next, a collection of chart paramterizations—encoders $\mathbf{E}_i$ as analogy of $\phi_\alpha$—map the initial latent representation to $N$ different chart spaces $U_\alpha$, which provides the new multi-chart latent space. Each chart representation is then passed into a decoding function, a decoder $\mathbf{D}_i$ as analogy of $\phi_\alpha^{-1}$, which produces approximation to the input data. Finally, the chart prediction module decides which chart and the associated decoder best represent the data. As detailed in A.3, this architecture, along with the proposed loss function, naturally enforces the chart transitions without explicit computation.

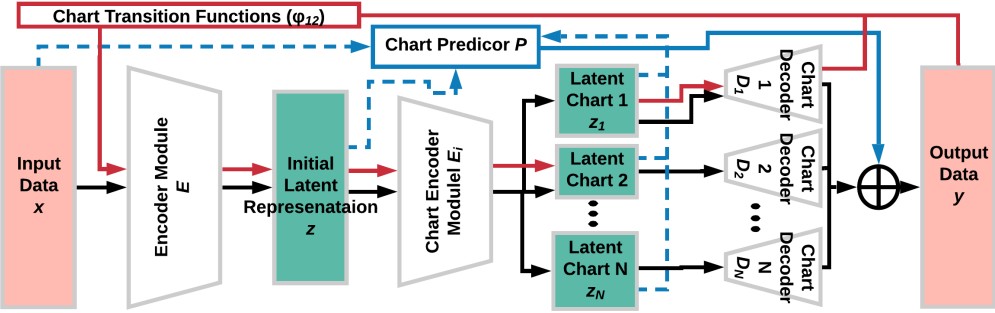

Figure 3: Architecture diagram of CAE and transition functions. The red path illustrates the computation of transition function $\phi_{12}$.

**Initial Encoder** The initial encoder serves as a dimension reduction step to find a low-dimensional *isometric* embedding of the data from $\mathbb{R}^n$ to $\mathbb{R}^l$. For example, given an $\mathbb{R}^3$ torus embedded in $\mathbb{R}^{1000}$, the initial encoder maps from $\mathbb{R}^{1000}$ to a lower dimensional space, ideally $\mathbb{R}^3$. (Note that however three is not the intrinsic dimension of the torus; rather, two is. Thus, subsequently we introduce chart encoders to map from the 3-dimensional space to $\mathbb{R}^2$.) We approximate this mapping using a neural net, denoted as $\mathbf{E}$, with a combination of fully connected and convolution layers (see section A.5 for details). We choose $l \ll n$; this encoding can be viewed as a dimension reduction step, which prepares the data to be split into each of the multi-chart latent spaces. Ideally, this step preserves

the original topology and geometry information of the data set while also reducing its dimension to the minimal isometric embedding dimension of the manifold. It aims at improving computational efficiency for decoders to multi-chart latent space. This step can be replaced with a Homeomoric Variation Auto-Encoder in the cases where the topology of the data set is known (Falorsi et al., 2018), or with an appropriately chosen random matrix projection (Baraniuk & Wakin, 2009; Cai et al., 2018).

**Chart Encoder**    This step provides parameterization of the data manifold locally to a chart space, whose dimension is approximately close to the intrinsic dimension of the data manifold. This splitting is done with a small collection of networks $\{\mathbf{E}_\alpha\}$ which takes $z \in \mathbb{R}^l$ as input and outputs $N$ local coordinates $z_\alpha \in U_\alpha$. We denote the direct sum of these space as $\mathcal{U} = \bigoplus_{\alpha=1}^N U_\alpha$, which is the proposed multi-chart latent space for our model. In practice, we choose $U_\alpha = [0, 1]^l$ and regularize the Lipschitz constant of the corresponding encoding map, to control size and regularity of the region $\mathcal{M}_\alpha \subset \mathcal{M}$ paramterized by $U_\alpha$ (more details in section 4).

We remark that the proposed multi-chart architecture aims at constructing the correct latent space structure and understanding the geometric structure of the data manifold. The decoupled nature of the encoding operations mean that the model tends to be larger in terms of the number of parameters. However, the improvement shown in the experiments is not caused by the use of more parameters; rather, a correct latent space structure. Further experiments in Appendix A.10 show that increasing the number of parameters in a VAE alone (without increasing the latent dimension) does not allow one to simultaneously produce good reconstruction and generation. A latent space of too small dimension will not be able to cover a manifold, and one of a too large dimension will generate points far from the manifold. Thus, the structure of the latent space is more important than the number of parameters.

**Decoders**    Each of the latent chart $U_\alpha$ is equipped with an associated decoder function $\mathbf{D}_\alpha$, which maps from the chart latent space $U_\alpha$ back to the ambient space. We represent each of these maps with a deep network, which are trained to reconstruct the input data.

**Chart Selection**    There are several options to the chart selection module $\mathbf{P}$ for an input $x$ sampled in the data manifold. In general, this module must produce prediction or confidence measure $\{p_\alpha(x)\}_\alpha = \mathbf{P}(x)$ regarding which chart should be used for a given input. After training, this module can also be used to reduce the computational cost of evaluating the model, by ensuring that only a small set of decoder functions needs to be evaluated for any input signal.

**Output**    The output of the network will depend on the application and the training specifics (discussed further in section 4.1). In general, the network produces an internal latent representation $z_\alpha(x)$ for an input $x$, a reconstruction signal $y \approx x$ to check the fidelity of the system, as well as some confidence $\{p_\alpha(x)\}$ in this prediction. Each of $y_\alpha = \mathbf{D}_\alpha \circ \mathbf{E}_\alpha \circ \mathbf{E}(x)$ may be used as a proxy for $y$ (and then each $p_\alpha$ can be interpreted as this probability) or some combination of the $y_\alpha$s may be used (in which case the $p_\alpha$ are interpreted as the partition of unity weights).

## 4    NETWORK TRAINING DETAILS

In this section, we explain several task specific modeling options, loss functions, regularization, and pre-training schemes that promote training efficiency of the proposed model.

### 4.1    CHART PREDICTION AND LOSS FUNCTIONS

The chart prediction module assigns any input to one or more charts. If the data manifold has relatively simple geometry, such a a circle, we may use the normalized distance from the data point to the center of the patch for prediction. This procedure is extremely efficient, but is not sufficiently powerful in cases where the geometry is more complex. For example, for the same surface area, a high curvature may require many small charts to cover, whereas a flat region may need only one chart. In this case we can compute $p_\alpha$ with a deep network, denoted as the chart predictor in Figure 3, using some combination of $x$, $z$ and/or $z_\alpha$ as inputs. Using $x$ as an input results in a network which is independent of the rest of the network (and can potentially be trained separately), but the

size and complexity of this network will depend on the ambient dimension of $x$. Using the internal representation $z$ or $z_\alpha$ as an input to this network allows the chart selection module to benefit from the dimension reduction of $x$ preformed by $\mathbf{E}$.

We propose two loss functions which lead to two slightly different interpretations of the model, based on how to handle regions in which the charts overlap. In the first regime, we define a decoder-wise loss for $y_\alpha = \mathbf{D}_\alpha \circ \mathbf{E}_\alpha \circ \mathbf{E}(x)$ as $e_\alpha = \|x - y_\alpha\|^2$ and an internal label $\ell_\alpha = \mathrm{softmax}(e_\alpha)$. Then the *Chart-Prediction Loss* is given by:

$$\mathcal{L}_{CP}(x, \Theta) := \left( \min_\alpha e_\alpha \right) - \sum_{\beta=1}^{N} \ell_\beta \log(p_\beta), \tag{2}$$

where $\Theta$ are network parameters. The first term models the reconstruction error of the predicted chart and the second term is the log-likelihood of the prediction, weighted by decoder-wise error.

The second regime is based on the idea of partition of unity idea (see Deng & Han (2008) for details). Here, we view $p_\alpha : \mathcal{M} \to \mathbb{R}$ as a function with compact support in $\mathcal{M}$, i.e. $\{x \in \mathcal{M} \mid p_\alpha(x) \neq 0\} \subset \mathcal{M}$, $p_\alpha \geq 0$ for all $\alpha$, and $\sum_\alpha p_\alpha = 1$. They serve as the partition of unity (See Figure 8 for a example). Thus, we represent any point on the manifold as a combination of the charts and use the confidence weights predicted by the chart predictor as the coefficients. The loss is then given by the following *Partition of Unity Loss*:

$$\mathcal{L}_{PoU}(x, \Theta) := \left\| x - \sum_{\alpha=1}^{N} p_\alpha y_\alpha \right\|^2 \tag{3}$$

## 4.2 REGULARIZATION

Since it is impossible to know *a priori* the number of charts necessary to cover a given data set, we instead overparameterize the model by using many charts and enforce a strong regularization on the decoder functions to eliminate unnecessary charts. Note that during training, a chart function (say $\mathbf{E}_j$) which is not utilized in the reconstruction of a point (i.e. $p_j \approx 0$) does not get any update from the loss function. Then, adding any convex penalty centered at 0 to the weights of $\mathbf{E}_j$ will result in weight decay and, if a decoder is never utilized during training, its weights will go to zero. In practice, we can automatically remove these charts by eliminating them from the network when the norm of the decoder weights falls bellow some tolerance. This mechanism provides a way of choosing the number of charts used in the network. Namely, we over estimate the number of charts and let the network automatically eliminate unnecessary ones, resulting in an appropriate number.

We also introduce an additional regularization to stabilize the training of our network by balancing the size of $\mathcal{M}_\alpha$ parameterized by $\mathbf{E}_\alpha$ and stopping a small number of charts from dominating the data manifold. For example, if we would like to use our network to model a (finitely sampled) sphere $S^2$, then we need at least two 2-dimensional charts. However, if we regularize a network with only $l_2$ weight decay, it may be able to reconstruct the training data well with only one chart without capturing the true structure of the data (see right panel of Figure 4 for such an example) . To prevent this type of overreach, we add a Lipschitz regularization to the decoders to penalize how far away nearby inputs can be mapped. Formally, the *Lipschitz constant* of a function is defined as: $\sup_{x \neq y} \frac{|f(y) - f(x)|}{|x - y|}$. Since the chart spaces are fixed as $[0, 1]^l$, controlling the Lipschitz constant of the encoder will control the max (euclidean) volume a chart can map onto. To do this, we note that the Lipschitz constant of a composition of functions can be upper bounded by the multiplication of the Lipschitz constants of its constituent functions. The Lipschitz constant of a matrix is its spectral norm and the Lipschitz constant of the ReLU is 1. Then, we can control the upper bound of the Lipschitz constant of an encoder function by regularizing multiplication of the spectral norm of its weights at each layer.

Combining these ideas, and denoting the weights of the $k^{th}$ layer of $E_\alpha$ as $W_\alpha^k$, we propose the following regularization on the decoder functions for a $K$-layer network:

$$\mathcal{R}_{Lip} := \max_\alpha \prod_{k=1}^{K} ||W_\alpha^k||_2 + \frac{1}{N} \sum_{\beta=1}^{N} \prod_{k=1}^{K} ||W_\beta^k||_2 \tag{4}$$

Here the first term aims at stopping a single chart from dominating, and the second term works as a weight decay which also promotes the smoothness of each chart.

### 4.3 PRE-TRAINING

Since our model jointly predicts the chart outputs and chart probabilities, it is important that our model is properly initialized, so that the range of each decoder lies somewhere on the manifold and the probability that a randomly sampled point lies in each chart is roughly equal. To do this, we begin by using the furthest point sampling (FPS) scheme to select $N$ data points, $\{x_\alpha\}$, from the training set which are 'far away' from each other. Then we assign each of these data points to a decoder and train each one to reconstruct. Additionally, we train the encoder such that $x_\alpha$ is at the center of the chart space $U_\alpha$. We further define the chart prediction probability as the categorical distribution and use it to pre-train the chart predictor. Then the loss for the $\beta^{th}$ initialization example is:

$$\mathcal{L}_{init}(x_\beta) := \|x_\beta - \mathbf{D}_\beta \circ \mathbf{E}_\beta \circ \mathbf{E}(x_\beta)\|^2 + \|\mathbf{E}_\beta \circ \mathbf{E}(x_\beta) - [.5]^d\|^2 + \sum_{\alpha=1}^{N} \delta_{\alpha\beta} \log(p_\alpha). \quad (5)$$

We can extend this idea of pre-training to also ensure that the charts are oriented consistently. Details are presented in A.4.

We remark that the pre-training network does not aim at separating the data manifold as different clusters. The pre-training works to ensure that each of the decoders is on the manifold, so that when training begins no decoders stay inactive. Since the chart selection module is learned in conjunction with the rest of the model, we do not assume any prior segmentation of the data. During training the charts will move, change sizes, overlap, or disappear.

## 5 NUMERICAL RESULTS

We present numerical results of several experiments to demonstrate the effectiveness of the proposed CAE. First, we study three illustrative geometric examples—sphere, double torus, and a genus-3 surface—to understand the behavior of the network. Afterward, we use the MINIST data set to further demonstrate the properties of CAE. In the end, we evaluate CAE and compare with other models on several metrics with the use of synethetic data sets as well as MNIST and SVHN.

### 5.1 ILLUSTRATIVE EXAMPLES

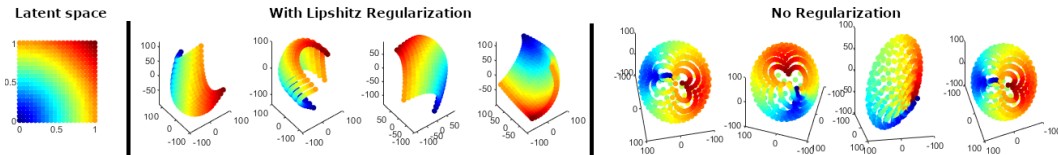

Figure 4: Left: Latent space. Middle: Model with Lipschitz Regularization. Right: Unregularized Model

In our first experiment illustrated in Figure 4, we visualize the process of applying a four-chart CAE to a data set sampled from the unit sphere (see Appendix A.2 for the network architecture). We apply the proposed loss function with and without the Lipschitz regularization discussed in section 4.2. We use four copies of $(0, 1)^2$ as the chart latent space in this experiment. We color code $(0, 1)^2$ using the distance of each point to the origin. After training, we uniformly sample points on the latent space and use the learned decoders to generate points back to the unit sphere. As we can see from the middle panel of Figure 4, the four charts, when glued together, successfully cover the unit sphere. Moreover, all charts occupy the data manifold in a balanced and regularized way; that is, even thought they are not uniform, no single chart dominates the rest. From the right panel of Figure 4, we can see that, when no regularization is employed, the charts are less localized. This behavior shows the necessity of using Lipschitz regularization to control the regularity of the decoder.

Our second experiment is conducted on a more complex object—the double torus—shown in Figure 1. The experiment illustrates some of the difficulties in using traditional auto-encoders to capture topologically complex data. Here, the data manifold has local dimension of 2, but it is not homeomorphic to a plane. We uniformly sample the latent space of each model and apply the resulting decoder to generate points back to the ambient space. As we can see from the second left plot in Figure 1, a traditional model with a 2-dimensional single-chart latent space cannot capture the overall manifold. Since this object can be embedded in $\mathbb{R}^3$, a model with a 3-dimensional latent space can capture the entire manifold. However, this type of model also likely generates points off the manifold, as we can see from the second right image in Figure 1. Finally, we see that our CAE with four 2-dimensional charts can produce points successfully covering the objects without introducing unfaithful points.

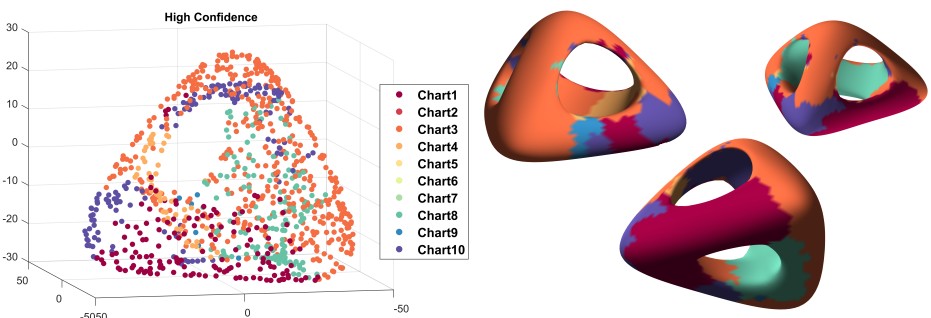

Figure 5: Left: Points sampled from high confidence regions. Right: Segmentation of manifold from chart selection

Next, we test our CAE on a genus-3 surface with ten 2-dimensional charts (detailed as CAE 2 in A.2). The left of Figure 5 shows the result of randomly sampling $z_\alpha$ in the chart latent space $\mathcal{U}$, and decoding the latent representations. The right of this figure shows which chart is active in each region of the manifold. Since this model uses a network to predict the chart segmentation, the resulting parameterization has charts of varying sizes. This allows the network to place more charts in areas of high curvature, and allow charts to grow over more linear regions. Nevertheless, this example demonstrates the effectiveness of our method to handle objects with complex topology.

## 5.2 THE MNIST MANIFOLD

We apply the 10-chart model on the MNIST data set (now using CAE 3 as detailed in A.2). The left panel of Figure 6 reports the reconstruction result in the training data, for a given image showed in the second last row. Each of the first ten rows in the corresponding column shows the decoding result from the $i$-th chart. Note that while each decoder may produce vastly different outputs, the chart selection module chooses which is most likely to be correct. As we can observe from the image, the chart selection model successfully picks up the most faithful decoding result, as we circle and repeat in the last row of the image. This picture shows that the proposed multi-chart auto-encoder does provide faithful reconstruction for the training data.

The middle panel of Figure 6 shows decoding results by sampling the charts, where each row shows images generated from the corresponding decoder. Note that each chart produces only a few digits, even though every digit is covered by some chart. Additionally, on any chart the digits which that chart produces are "close" to each other (for example the 3s and 8s in chart 8 and the 5s and 6s in chart 1). This means the multi-chart latent space can cover the MINST data manifold in a balanced and regular way, similar to what we observe from the experiments conducted for geometric objects. The right panel of this figure shows the morphing of a '2' to a '3' by interpolating a linear path through the latent space. Since each of the latent representations decoded along this path produces output similar to examples found in the training set, we can conclude that approximation of the manifold given by our chart parameterization is close to the underlying distribution the training data is sampled from. In traditional approaches this is possible because the latent space (modeled as a normal distribution) is simply connected. Our model is able to do so without using a distributional assumptions, owing to the transition conditions and Lipshitz regularization.

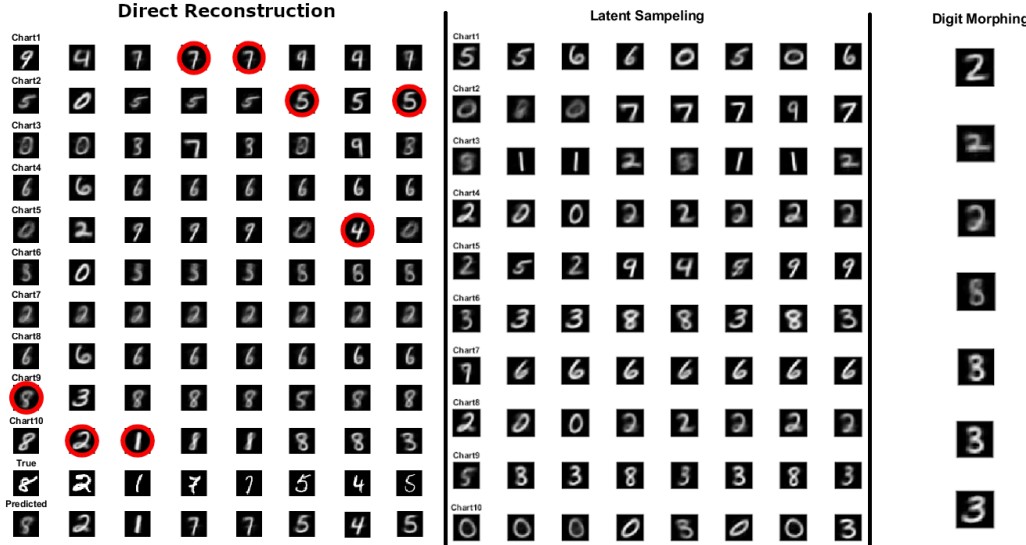

Figure 6: Left: Reconstruction of data from MNIST data set. Middle: Reconstruction from random sample on the multi-chart latent space. Right: Digit morphing

## 5.3 MODEL EVALUATION

In this experiment, we apply four traditional models (2 auto-encoders and 2 variational auto-encoders) as well as three CAEs on several data sets. Details of the exact architecture of these networks can be found in A.2. For each model and data set, we are primarily interested in three measures of success, including reconstruction error, unfaithfulness, and coverage (See A.10 for detailed definitions). The reconstruction error measures the fidelity of the model. The unfaithfulness measures how far synthesized data decoded from samples drawn on the latent space are to samples from the original training data. Coverage indicates how much of the training data is covered by the encoder. Models which produce unrealistic data when sampling from the latent space will have high unfaithfulness sores and models which experience mode collapse will have low coverage scores.

We test these measurements on four data sets, in the order from the simplest to the most complex. ① **Sphere**: The data consists of 2000 equally distributed points sampled uniformly form a sphere embedded in $\mathbb{R}^3$. ② **Genus 3**: The genus-3 object used in Figure 5 non-trivally embedded in $\mathbb{R}^{50}$. ③ **MNIST**: The MNIST hand-written digits database containing 60k training and 10k testing images (LeCun et al., 1990). ④ **SVHN**: A real-world image dataset from house numbers in Google Street View images. We focus on the individual digits problem and preprocess the images in gray scale (Netzer et al., 2011). Results of the the evaluation metrics are summarized in Figure 7 and reported fully in Table A.10 in A.10.

From these results, clearly the CAE models consistently preform better than other models with simple latent spaces. More specifically, when the dimension of the latent space is fixed, the CAE model preforms better than the associated VAE and AE in every test. Moreover, because of the Lipschitz regularization in our model, we are able to achieve much better coverage results than with the previous methods.

## 6 CONCLUSIONS AND FUTURE WORK

We have proposed and investigated the use of chart-based paramterization to model manifold structured data, through introducing multiple-chart latent spaces, along with transition functions, to autoencoders. The parameterization allows us to significantly reduce the dimension of latent encoding for efficiently representing data with complex structures. Numerically, we design geometric examples to analyze the behavior of the proposed CAE and illustrate its advantage over single-chart

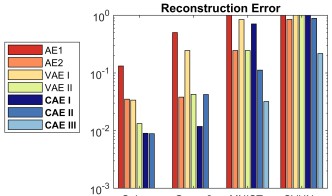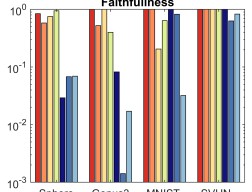

Figure 7: Summary of benchmark test on Sphere, Genus-3, MNIST and SVHN data sets

autoencoders. We also apply our method to real-life data sets, including MNIST and SVHN, to demonstrate the effectiveness of the proposed model.

We believe that the proposed chart-based parameterization of manifold-structured data provides many opportunities for further analysis and applications. In future work, we will extend this architecture to other generative models (e.g, GAN) and apply the machinery to investigate the topology and geometry of real-world data.

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

# A  APPENDIX

## A.1  NOTATION

**Objects:**
Data: $x \in D \in \mathbb{R}^n$
Chart Center: $c_i \in D$ Dataset: $D$
Manifold: $\mathcal{M} \subset \mathbb{R}^n$
Latent Representation: $z \in \mathbb{R}^l$
Chart Representations: $z_\alpha \in \mathbb{R}^l$
Chart Space: $U_\alpha := [0,1] \in \mathbb{R}^l$
Discretized chart Space: $\bar{U}_\alpha := [0,1]^l \in \mathbb{R}^l$
Chart Prediction: $p \in \mathcal{P}(N)$
Reconstructed Data: $y \in \mathbb{R}^n$
Residual $e_i \in \mathbb{R}$
Discretization of chart: $\{b_i^j\}_j \subset U_\alpha$
Parameters: $\theta$
Internal Label : $l_i$

**Functions:**
Initial Encoder: $\mathbf{E} : x \to z$
chart Encoder: $\mathbf{E}_i : z \to z_\alpha$
chart Decoder: $\mathbf{D}_i : z_\alpha \to y_i$
chart Predictor: $\mathbf{P} : (z_\alpha; z; x) \to p$
chart Transition: $\mathbf{T}_{ij} : z_\alpha \to z_j$
Chart Function: $\phi_i : U_\alpha \to \mathcal{M}$
Function on Manifold: $f : x \to \mathbb{R}$
Loss Functions: $\mathcal{L}_{name}$
Regularization: $\mathcal{R}_{name}$
Error Measurement: $\mathcal{E}_{name}$
Density function: $\eta : (y_i; z_\alpha; x) \to [0,1]$
PCA Projection: $U_\alpha \to \mathbb{R}^d$

## A.2 MODEL DETAILS

In this section we detail the architecture of the networks used in the numerical experiments. We denote fully connected layers as $FC_y$ where $y$ is the number of units in the layers, $Conv_{i,j,k,l}$ as convolution layers with filters of size $(i, j)$ input dimension $k$ and output dimension $l$, and $dim(U)$ to be the dimension of the latent space. Each model was trained using the chart prediction loss function as we have found it to be more stable during training.

**Auto-Encoders**:

$$\begin{aligned} \text{Encoder} &: x \to FC_{250} \to FC_{250} \to FC_{250} \to FC_{dim(U)} \to z \\ \text{Decoder} &: z \to FC_{250} \to FC_{250} \to FC_{250} \to FC_N \to y \end{aligned} \tag{6}$$

**Variational Auto-Encoders**:

$$\begin{aligned} \text{Encoder} &: x \to FC_{250} \to FC_{250} \to FC_{250} \to FC_{2*dim(U)} \to z, \sigma \\ \text{Decoder} &: z \in \mathcal{N}(z, \sigma) \to FC_{250} \to FC_{250} \to FC_{250} \to FC_n \to y \end{aligned} \tag{7}$$

**CAE 1 (4 2-dim charts, distance confidence, chart predictor)**:

$$\begin{aligned} \text{Initial Encoder} &: x \to FC_{250} \to FC_{250} \to FC_{250} \to z \\ \text{Chart Encoder} &: z \to FC_{250} \to FC_{dim(U_i)} \to z_i \\ \text{Decoder} &: z_i \in \mathcal{N}(z, \sigma) \to FC_{250} \to FC_{250} \to FC_{250} \to FC_n \to y \\ \text{chart Prediction} &: softmax(||z_i||) \end{aligned} \tag{8}$$

**CAE 2 (10 2-dim charts, learned confidence chart predictor)**

$$\begin{aligned} \text{Initial Encoder} &: x \to FC_{250} \to FC_{250} \to FC_{250} \to z \\ \text{Chart Encoder} &: z \to FC_{250} \to FC_{dim(U_i)} \to z_i \\ \text{Decoder} &: z_i \in \mathcal{N}(z, \sigma) \to FC_{250} \to FC_{250} \to FC_{250} \to FC_n \to y \\ \text{chart Prediction} &: x \to FC_{250} \to FC_{10} \to softmax \to p \end{aligned} \tag{9}$$

**CAE 3 (10 25-dim charts, convolution layers, learned chart predictor)**

$$\begin{aligned} \text{Initial Encoder} &: x \to Conv_{4,4,1,4} \to Conv_{3,3,4,8} \to Conv_{2,2,8,16} \to z \\ \text{Chart Encoder} &: z \to FC_{250} \to FC_{dim(U_i)} \to z_i \\ \text{Decoder} &: z_i \in \mathcal{N}(z, \sigma) \to FC_{250} \to FC_{250} \to FC_{250} \to FC_n \to y \\ \text{chart Prediction} &: z \to FC_{250} \to FC_{10} \to softmax \to p \end{aligned} \tag{10}$$

## A.3 CHART TRANSITION FUNCTIONS

A key feature of the chart-based parameterization of manifolds in differential geometry is the construction of chart transition functions. As show in figure 2, some points on the manifold may be parameterized by multiple charts. Let $\phi_\alpha$ and $\phi_\beta$ be two charts with overlapping domains $(M_\alpha \cap M_\beta \neq \emptyset)$, then the chart transition function $\phi_{\alpha\beta}$ can be computed as: $\phi_\alpha^{-1}\phi_\beta$. In our model the $\phi^{-1}$s are represented as $E_i$ neural networks, but directly computing $\phi$ itself from $\phi^{-1}$ is not simple since ReLU-nets are non-invertable. It would be possible to add additional modules and to train a model to predict the transition function, but this adds up to $\binom{N}{2}$ new networks to train, many of which may be unnecessary (since we only need chart transition function for overlapping charts). However, we can exploit the structure the encoder module and re-encode the signal generated by the first decoder, using the second encoder define a chart transition. Then Each chart transition function can be modeled by the composition:

$$\phi_{\alpha\beta} : U_\alpha \cap U_\beta \to U_\beta \cap U_\alpha, \qquad z_\alpha \mapsto \mathbf{E}_\beta\Big(\mathbf{E}\big(\mathbf{D}_\alpha(z_\alpha)\big)\Big) \tag{11}$$

Note that if $x \in \mathcal{M}_\alpha \cup \mathcal{M}_\beta$, then to estimate the chart transition between $U_\alpha$ and $U_\beta$ we need: ① $p_\alpha(x) \approx p_\beta(x)$ ② $x \approx \mathbf{D}_\alpha(\mathbf{E}_\alpha(\mathbf{E}(x)))$ ③ $x \approx \mathbf{D}_\beta(\mathbf{E}_\beta(\mathbf{E}(x)))$ Each of these conditions

are naturally enforced by both of loss functions equation 2, equation 3 discussed in the precious section. Therefore the chart transition function of this network can be computed without explicitly parameterizing them or adding new terms to the loss function. One could explicitly characterize the the chart transition by re-encoding the decoded signals in a second pass thought the network and computing an regularizer:

$$\mathcal{R}_{cycle}(x) := \|x - \mathbf{D}_\beta \circ \mathbf{E}_\beta \circ \mathbf{E} \circ \mathbf{D}_\alpha \circ \mathbf{E}_\alpha \circ \mathbf{E}(x)\| + \|x - \mathbf{D}_\alpha \circ \mathbf{E}_\alpha \circ \mathbf{E} \circ \mathbf{D}_\beta \circ \mathbf{E}_\beta \circ \mathbf{E}(x)\| \quad (12)$$

which measures the exactly error in the transition and reconstruction. However this type of cyclic condition is computationally expensive to implement, and we have found it unnecessary in our empiric studies.

## A.4 PCA CHART ORIENTATION

We can further extend the idea of pre-training to also orient the rest of the chart around the center $c_\alpha$. To do so, we take a small sample of points $\mathcal{N}(c_\alpha)$ around the center and use principle component analysis (PCA) to define a $d$-dimensional embedding of the local neighborhood. Let the coordinates of this neighborhood embedding be: $\hat{x}_\alpha(x) := \frac{1}{C_\alpha} W_\alpha x + b_\alpha$ for $x \in \mathcal{N}(c_\alpha)$ where $W_\alpha$ is the optimal orthogonal projection from $U_\alpha$ to $\mathbb{R}^d$ and $b_\alpha$ shifts $\hat{x}_\alpha(c_\alpha)$ to $[.5]^d$ and $C_i$ is chosen as a local scaling constant. Then we can use this coordinate system to initialize the orientation of the local charts by adding an additional regularization the term to the equation 5:

$$\mathcal{R}_{cords} = \sum_{\alpha=1}^{N} \sum_{x \in \mathcal{N}(c_\alpha)} \langle \mathbf{E}_\alpha(\mathbf{E}(x), \hat{x}_\alpha(x)) \rangle \quad (13)$$

## A.5 CONVOLUTION

It has been widely argued that invariant and equivalent properties of convolutions layers promote manifold structured representations. For example, (Shaham et al., 2018) conjectures: 'that in a representation obtained as an output of convolutional and pooling layers, the data concentrates near a collection of low-dimensional manifolds embedded in a high-dimensional space.' In other words, applying dimension reduction operations which has localized invariance (such as convolution and pooling) maps data to relatively simple manifolds by contracting the representation space. This suggests that adding convolution and pooling layers to the beginning of the encoder networks will result in representations which are easier for our model to estimate since the underlying geometry will be simpler.

## A.6 NETWORK QUALITY MEASUREMENTS

We write $\mathbf{D}$ as a general notation for the decoder in the model.

**Reconstruction Error**   Measures fidelity of the output $y$: $\mathcal{E}_{recon} = \frac{1}{|D_{test}|} \sum_{x \in D_{test}} ||x - \mathbf{D}(x)||^2$.

**faithfulness Error**   Measures how close data decoded from samples drawn on the latent space are to samples from the original training data. We uniformly selecting $M$ points $\{z_i\}_{i=1}^{M}$ in the latent space and define $\mathcal{E}_{sample} = \frac{1}{M} \sum_{i=1}^{M} \min_{x \in D_{train}} \|x - \mathbf{D}(z_i)\|^2$. Often, people hope that through sampling the latent space and decoding, one can get new data in the original space that *novel* or *original*. However, if the training set is sufficiently dense on the data manifold, newly generated data is far from anything observed during training will not be realistic. Since we are concerned with fitting a model to the data, a properly trained model should always stay close the the underlying manifold. To select the latent variables used in this experiment, we uniformly draw $M = 100$ samples from the latent space used in the single chart auto-encoder, CAE and VAE.

**Coverage**   Measures how well the latent space and decoders capture the entire data set. We uniformly draw $M$ samples $\{z_i\}_{i=1}^{M}$ from the latent space. Let $M^*$ be the number of the set $\{x^* \mid x^* = \arg\min_{x \in D_{train}} \|x - \mathbf{D}(z_i)\|^2 \text{ for some } i\}$. We define $\mathcal{E}_{coverage} = \frac{M^*}{M}$. This measurement provides quantitative way to describe the well known "mode collapse" problem of GAN

(Arjovsky et al., 2017) wherein some the model captures part of the data well, but ignores large sections (or entire classes) of the data. A coverage score close to one indicates that the samples are well distributed on the manifold, while scores close to zero indicate that the model may be experiencing mode collapse.

## A.7 CHART TRANSITIONS

In this experiment, we train a model with four 1-dimensional charts to fit a circle in order to visualize the transition between charts. In figure 8, the first row shows the output of each chart using the latent variable $z_i$ sampled on $(0, 1)$. The top right shows the chart which has the largest $p$ value. In the second row, we visualize the transition zones. Here, the solid colored lines are decoded samples taken from the $U_i$ space. The '+'s represent training data, and their color indicates which chart had maximum $p$ value. The ground truth manifold is represented by the dashed lined. The last row show the partition of unity function, unwrapped from the circle to a line. From this experiment we can see that charts have very close values in these transition zones and the partition of unity functions are compactly supported.

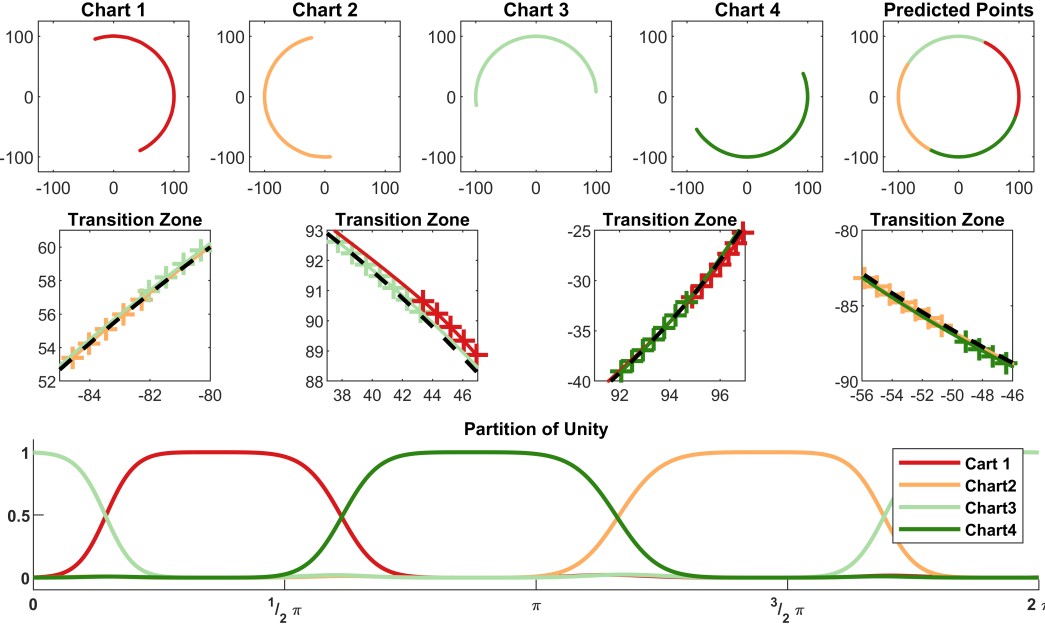

Figure 8: Top row: Embedding of Charts and Maximum of Patch Prediction. Second row: chart transitions. Training data is marked with '+' and samples taken from the charts are denoted by the solid colored line. The black dash line shows the ground truth. Bottom: Partition of Unity functions

## A.8 AUTOMATIC CHART REMOVAL

In this experiment, we train a model with four 2-dimensional charts for data sampled on a sphere in order to visualize the effect of the regularization scheme when using a neural network as the chart prediction module. Figure 9 shows the patch prediction function on the training data at the end of the pre-training and at the completion of training, respectively. From this figure we see that even though all charts cover the sphere at the beginning, the regularization is able to automatically remove some charts after training.

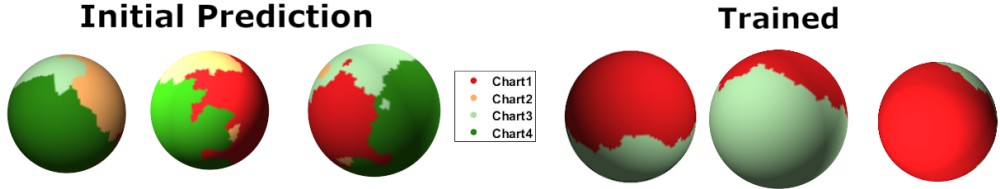

Figure 9: Left: Initial Predictions . Right: Final regions

### A.9 MEASURING GEODESICS

In this experiment, we demonstrate a simple example of recovering geometric information from a trained model by computing the length of some geodesic curves on a sphere. Let $p$ and $q$ be points on the manifold with latent representations $z_p$ and $z_q$. Then the line $\gamma_z(t) = \alpha z_p + (1 - \alpha) z_q$ in the latent space will correspond to a path on $\gamma \subset \mathcal{M}$. To measure the length of this path, we can sample points along $\gamma_z(t)$, decode them and then measure the euclidean distance between the decoded points. Figure 10 shows an example of such a test using different numbers of sampling points for five difference curves on the sphere. From this experiment we observe convergence of these measurements as more points are sampled along the geodesic path, validating our geometric intuition. We remark that this is a very preliminary result to show a potential of understanding geometric structure of data manifold using multi-chart latent space. We will explore in the direction in our future work.

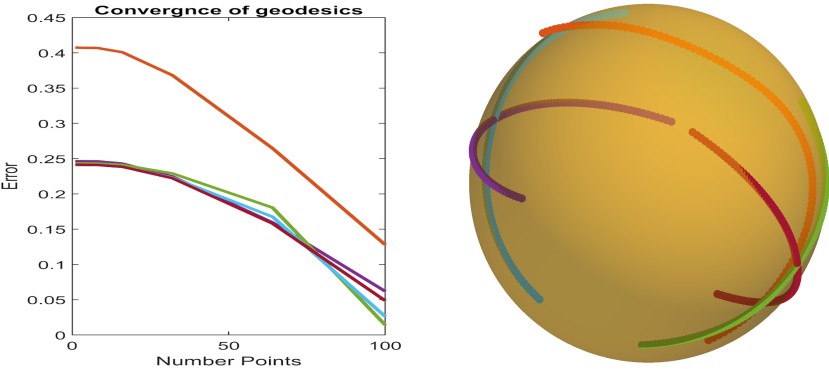

Figure 10: Left: Geodesic approximation error v.s. number of points sampled in the latent space. Right: Geodesic curves generated from the decoder network.

### A.10 DETAILED BENCHMARK COMPARISONS

Figure 11 shows an experiment for data sampled on a double torus using VAE with different choice of parameters. The latent space dimension is chosed as 2 which is compatible with the intrinsic diemisno of the object. This experiment shows that increasing the number of parameters in a VAE alone (without increasing the latent dimension) does not allow one to simultaneously produce good reconstruction and generation. A latent space which has too small of a dimension will not be able to cover a manifold, and one which is too large will generate points far from the data manifold. Thus the structure of latent space is more important than the number of parameters. This is one of the main objectives of this paper.

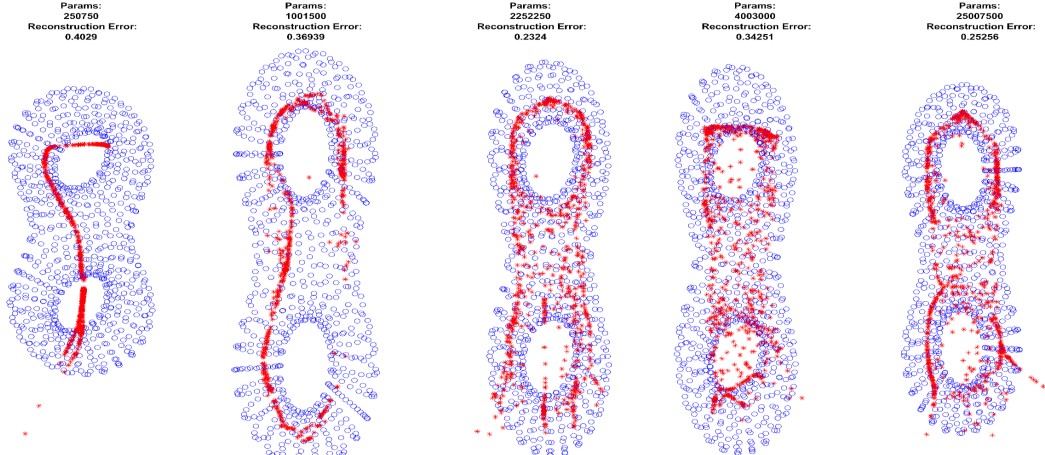

Figure 11: Five overparametized VAE with 2 dimensional latent space for data sampled double torus. Blue:training data, Red:samples from latent space.

| Model (Latent Space) | # of Param. | Data | Intrinsic /Amb. | | Recon. Error | Unfaithfulness | Coverage |
|---|---|---|---|---|---|---|---|
| AE I $(\mathcal{U} = [0,1]^2)$ | 128503 | Sphere | 2 | 3 | 1.32e-01 | 8.409e-01 | .85 |
| | 152050 | Genus 3 | 2 | 50 | 5.018e-01 | 3.286e00 | .15 |
| | 519784 | MNIST | ? | 784 | 3.111e00 | 3.486e00 | .18 |
| | 640024 | SVHN | ? | 1024 | 4.235e00 | 8.123e00 | .12 |
| AE II $\mathcal{U} = [0,1]^{25}$ | 140003 | Sphere | 2 | 3 | 3.521e-02 | 5.789e-01 | .87 |
| | 163550 | Genus 3 | 2 | 50 | 3.819e-02 | 5.233e-01 | .68 |
| | 531284 | MNIST | ? | 784 | 2.459e-01 | 3.975e00 | .23 |
| | 651524 | SVHN | ? | 1024 | 8.523e-01 | 2.1756e00 | .14 |
| VAE I $\mathcal{U} = \mathcal{N}(0,I)^2$ | 129005 | Sphere | 2 | 3 | 3.38e-02 | 7.489e-01 | .88 |
| | 152552 | Genus 3 | 2 | 50 | 2.459e-01 | 2.575e-0 | .67 |
| | 520286 | MNIST | ? | 784 | 8.459e-01 | 2.045e-01 | .55 |
| | 57799 | | ? | 1024 | 6.137e00 | 8.100e00 | .43 |
| VAE II $\mathcal{U} = \mathcal{N}(0,I)^{25}$ | 146278 | Sphere | 2 | 3 | 1.33e-02 | 9.451e-01 | .92 |
| | 169825 | Genus 3 | 2 | 50 | 4.273e-02 | 4.002e-01 | .66 |
| | 537559 | MNIST | ? | 784 | 2.451e-01 | 6.377e-01 | .82 |
| | 657799 | SVHN | ? | 1024 | 4.012e-00 | 4.012e-00 | .75 |
| 4-chart Model $\mathcal{U} = \bigoplus_{i=1}^{4}[0,1]^2$ | 333266 | Sphere | 2 | 3 | 8.025e-03 | 2.921e-02 | 1 |
| | 392016 | Genus 3 | 2 | 50 | 1.187e-02 | 8.234e-02 | .86 |
| | 1309516 | MNIST | ? | 784 | 7.100e-01 | 8.100e00 | .57 |
| | 1707450 | SVHN | ? | 1024 | 9.0234e-00 | 8.904e00 | .63 |
| 10-chart Model $\mathcal{U} = \bigoplus_{i=1}^{10}[0,1]^2$ | 751790 | Sphere | 2 | 3 | 9.878e-03 | 6.785e-02 | 1 |
| | 881040 | Genus 3 | 2 | 50 | 4.245e-02 | 1.412e-01 | .82 |
| | 2899540 | MNIST | ? | 784 | 1.120e-01 | 8.231e-01 | .91 |
| | 3795750 | SVHN | ? | 1024 | 8.877e-01 | 6.311e-01 | .75 |
| 10-chart Conv. Model $\mathcal{U} = \bigoplus_{i=1}^{10}[0,1]^{25}$ | 3498564 | MNIST | ? | 784 | 8.556e-02 | 3.209e-02 | .87 |
| | 4098564 | SVHN | ? | 1024 | 2.177e-01 | 8.271e-01 | .88 |

