# OpenReview forum: "Chart Auto-Encoders for Manifold Structured  Data"
_ICLR.cc/2020/Conference — Reject_

### Official Review · AnonReviewer2 · 2019-10-09
**Official Blind Review #2**

**Rating:** 6

**Review:**

Notes:

  -Goal is to learn autoencoders which can capture disconnected manifolds by employing multiple discrete charts.

  -For example in Figure 1, a Torus is shown, which can't be captured by a single chart (smooth mapping from euclidean space).

  -This motivating intuition makes sense, although I wonder to what extent a neural network can compensate for this by being relatively unsmooth.

  -The circle example in the introduction is illuminating and I enjoyed it.

  -It's somewhat subjective, but I feel that autoencoders are becoming less widely used, so the paper might have more impact if it had targeted models like ALI/BiGAN which do reconstruction but purely with adversarial objectives.

  -Two techniques are presented for handling how points are assigned to charts (4.1).  I'm a bit unclear on how this interacts with the notion that a single point can be covered by multiple charts.

  -The paper uses lipschitz regularization on the decoder.  Note that Spectral Normalization (Miyato 2018) could also be used here.

  -The illustration of the effect of lipschitz regularization in figure 4 is good.

  -For 5.2, I'd prefer the use of a dataset other than MNIST, since we don't strictly know that the digits require different charts (for example I'm pretty sure there's a smooth mapping between "1" and "7").  I'd prefer an example where literally different types of objects are combined which couldn't possibly be modeled by a single chart.

Review: Overall I felt like this paper gives a nice mathematical exposition on the relationship between charts, manifolds, and autoencoders.  I slightly lean for acceptance but am very borderline, especially as the results on "real data" are very weak.

**Experience Assessment:**

I have published in this field for several years.

**Review Assessment: Checking Correctness Of Derivations And Theory:**

I assessed the sensibility of the derivations and theory.

**Review Assessment: Checking Correctness Of Experiments:**

I assessed the sensibility of the experiments.

**Review Assessment: Thoroughness In Paper Reading:**

I made a quick assessment of this paper.

---

> ### Author Response · Authors · 2019-11-15
> **Response to reviewer 2**
>
> 1. It's somewhat subjective, but I feel that autoencoders are becoming less widely used, so the paper might have more impact if it had targeted models like ALI/BiGAN which do reconstruction but purely with adversarial objectives.
>
> Response: We agree that adversarial models are a very important aspect of this type of work and further conjecture that it would be possible to train our models with such a loss. However, our primary focus in this paper is to study the chart structured latent space. To do this, it is very convenient to have an encoder  which operates as the approximate inverse of the generative process so that we can train the chart transition functions with a single forward and backward pass. We would like to explore GAN in the context of charts in future work.
>
> 2. Two techniques are presented for handling how points are assigned to charts (4.1). I'm a bit unclear on how this interacts with the notion that a single point can be covered by multiple charts.
>
> Response: We have presented two models: In the first, each input is assigned to a probability distribution over charts. For example, if the model predicts that p_i = p_j = ½ then we would say that this point is in the intersection of chart i and j. In the second model, p is modeled as the coefficients of the convex combination of the outputs of each decoder. Then the weights of the combination act as a soft assignment of the point into each of the decoders, this can be viewed as being covered by multiple charts.
>
> 3. The paper uses lipschitz regularization on the decoder. Note that Spectral Normalization (Miyato 2018) could also be used here
>
> Response: Thank you for this reference, as we are using the Lipsitz constant to bound the spectral norm, this would give us a tighter bound, although training with this constraint may be more time consuming.
>
> 4. For 5.2, I'd prefer the use of a dataset other than MNIST, since we don't strictly know that the digits require different charts (for example I'm pretty sure there's a smooth mapping between "1" and "7"). I'd prefer an example where literally different types of objects are combined which couldn't possibly be modeled by a single chart.
>
> Response: In general, we can view any traditional auto-encoder or variational auto-encoder as a single chart model without any transition conditions. Then by increasing the dimension of the latent space,,we could cover any data set with a sufficiently large latent space. If the dimension of the latent space is fixed, then there are examples of manifolds which cannot be captured by a single chart. Figure 1 shows an example of this: there is no map between a 2D Euclidean plane and the double torus. When trying to parameterize this data, the model with the 2D latent space fails to capture any of the ‘backside’ of the shape.

---

### Official Review · AnonReviewer1 · 2019-10-19
**Official Blind Review #1**

**Rating:** 3

**Review:**

## Summary of the Paper

This paper introduces a new architecture for autoencoders based on the
concept of *charts* from (differential) topology. Instead of learning a
single latent representation, the paper proposes learning charts, which
serve as local latent representations. Experiments demonstrate that the
local representations perform favourably in terms of approximating  the
underlying manifold.

## Summary of the Review

This is an interesting paper with an original idea. I appreciate the use
of concepts from differential topology in deep learning and agree with
the paper that such a perspective is required to increase our
understanding of complicated manifold data sets. However, I find the
following issues with the paper in its current form, which prevent me
from endorsing it for acceptance:

1. I have doubts about the technical correctness of the proposed
   architecture; specifically, the relevance of the *initial* latent
   representation, which employs a Euclidean space, is not analysed.

2. The role of the number of charts, which needs to be specified
   before-hand, is not analysed in an ablation study.

3. The experiments do not showcase the *conceptual* improvements of the
   proposed technique.

I shall briefly comment on each of these points before discussing other
improvements.

I want to point out that I really like the ideas presented in this
paper and I think it has the potential to make a strong contribution but
the issues in their current form require substantial revisions and
additions.

# Concern 1: Technical correctness

The paper claims at multiple places that the geometry of Euclidean space
is 'trivial' or 'too simplistic' to meaningfully reflect the structure
of the data. This claim is double-edged, though: first, there are
many methods that use autoencoders based on these spaces that exhibit
sufficient reconstruction capabilities. Second, the proposed
architecture itself uses a Euclidean latent representation as its
initial encoder. The paper states that 'Ideally, this step preserves the
topology of the data [...]', but this is never analysed.

I fully agree with the idea that charts are a suitable way to describe
complicated manifolds, but the paper needs more precision when terms
such as 'topology' and 'geometry' are being used. Likewise, I disagree
with referring to Euclidean space as 'trivial'. Again, other methods
demonstrate that the space captures high-level phenomena sufficiently
well for reconstruction purposes. At the very least, the paper should
be more precise here.

Moreover, I would recommend experiments in which the dimensionality of
the initial encoder is discussed.

# Concern 2: Number of charts

Selecting the number of charts appears to me as a critical component of
the proposed method. While the appendix contains one experiment for
MNIST with different numbers of charts, this concept needs to be fleshed
out more. How do we know that we have a sufficient number of charts?
Since in differential topology, the choice of chart should not matter,
how does it behave in these cases? Is there a way to detect that the
number of charts must be increased?

I could envision something like a simple 'step size' control procedure:
if a quality measure indicates that there need to be more charts, double
the number of charts and re-run the training; if the number of charts
is too big, halve it and re-run the training.

I get the idea that increasing the number of charts will probably
decrease the reconstruction error, but this comes at the obvious expense
of even more parameters. I thus recommend another set of experiments
that shows the influence of the number of charts, maybe even on the
synthetic data sets used in the paper.

# Concern 3: Conceptual improvements

While I enjoyed the didactic approach of the paper, which first
introduces simple test data sets to illustrate the concepts, my
main question is about the conceptual improvements that the charts
provide in the end.

I see that the reconstruction error for MNIST goes down---but there are
also significantly more (!) parameters than in the comparison
architectures. The ideas of the sampling or interpolation experiments
go in the right direction, but in their present version, they are not
entirely convincing. In fact, they even raised more questions for me:

- Figure 6 depicts individual charts but their *covering* of the space
  is highly non-uniform. The letter '0' is covered more often than the
  letter '1', for example. How can this be compatible with the claim
  that the novel architecture learns a suitable set of charts? I could
  understand some overlaps, but there seems to be a clear difference
  between the charts generated in the synthetic examples---which do
  appear to be cover everything in a uniform manner---and the charts for
  MNIST. This needs to be elucidated some more, in particular since the
  paper writes that the charts 'cover [...] in a balanced and regular
  way'.

- The digit morphing example is not not entirely convincing to me. Is
  this not something that I can do equally well with a VAE or generative
  models in general? I am *not* disputing the claims of the paper here,
  I am merely stating that *if* the new method is beneficial for this
  sort of application, a more in-depth experiment is required.

Thus, while I would like to give the paper the benefit of the doubt, it
does not show just *why* it is relevant to have a chart-based embedding.
Some suggestions for a set of experiments:

- Do charts help in separating the input space? I would hypothesise
  that this is the case---it thus might be worthwhile to study
  low-dimensional embeddings obtained based on each chart and 'stitch'
  them together.

- Do charts tell us something about the properties of a manifold? For
  example, are certain charts 'easier' to embed than others? This could
  be used to indicate different dimensions in a data set.

## Experimental setup

I have one major point of critique here, namely the way results are
presented without any measures of tendency. Instead of showing a bar
plot in Figure 7, I would suggest showing a Table with standard
deviations along multiple repetitions of the experiment. It is not clear
from looking at this to what extent this results can be replicated.

Moreover, a discussion of the number of parameters is required. To some
extent, I find it not surprising that a better reconstruction error is
achieved if more parameters are present.

This makes some of the claims in the paper hard to assess.

## Technical clarity

The papers is generally written well and has a good expository style.
Here are some cases where I find that clarity can be improved:

- To add to what I wrote above: if charts are Euclidean as well, the
  paper should elucidate why Euclidean charts do *not* suffer from being
  too simplistic.

- The discussion of homeomorphisms in the introduction is slightly
  misleading; none of the functions learned later on is a homeomorphism
  because of the latent space dimensions.

- Homeomorphic mappings of manifolds into a Euclidean space are not
  necessarily desirable---this is why the definition of a manifold uses
  the concept of neighbourhoods. I think this should be rephrased in
  a positive manner, as in: manifolds are complex, so we cannot expect
  a *single* map to suffice...

- The leading example of a torus embedding needs more details. Why is
  the structure destroyed?

- The introduction of 'topological features' on p. 2 is slightly abrupt.
  It would be sufficient to explain by means of the figure that the
  mapping obviously does not respect all properties.

- p.2: paths become invariant to _what_ exactly?

- p.2: what is the 'topological class'?

- p.2: would LLE not be a good precursor to the method proposed in this
  paper?

- p.2: is this paper to be seen as an implementation of Chen et al.  (2019)?
  This should be made more clear.

- p.3: the concept of intrinsic dimension slightly varies in literature.
  I would propose mentioning the homeomorphism of every chart to some
  $d$-dimensional space, and state that if this exists, one calls the
  manifold $d$-dimensional.

- p.3: the circle example could be explained in more detail for readers
  unfamiliar with the concepts.

- p.4: the chart prediction module requires a brief explanation at the
  point when it is first introduced (1 sentence is sufficient). The
  method plus architecture is presented but the details come very late;
  I would prefer some intuition here

- p.4: $N$ needs to be defined earlier

- p.4: how is the dimension of latent spaces chosen? Please also refer
  to my comments on the experiments above.

- p.5: Section 4.1 again mixes 'topological' and 'geometrical' concepts;
  suddenly, the concept of curvature crops up---this needs to be
  explained better!

- p.5: Distances can always be measured in connected subsets of
  real-valued spaces; whether the set is open or closed does not change
  the fact that a centre exists. Am I misunderstanding this?

- p.5: I like the 'partition of unity' approach, but to me, this reads
  like a convex combination of predictions. Am I misreading this? If
  not, I would suggest to rephrase this.

- p.5/6: the goals of the new method need to be stated more clearly; the
  paper needs to explain better to what extent *reconstruction error* is
  affected by charts (it does not seem to be, as I outlined above)---and
  this again raises the question of which quality measure the new method
  *can* preserve.

- p.6: the definition of the Lipschitz constant could be more precise;
  please specify the requirements $f$ has to satisfy

- Eq. 4 needs more details for me: it seems as if the weights appear
  twice as a kind of 'decay term' (in the second part, I see the sum
  but the product appears in both terms). This should be stated more
  clearly.

- p.6: the pre-training needs more details; how crucial is this step?

- p.6: what does the 'orientation' imply? It is not defined except in the
  appendix.

- p.6: the jump from the illustrative examples to the non-synthetic ones
  is large; the uniform sampling of the latent space does not scale to
  higher dimensions, for example. The paper should comment on this if
  possible.

- In general, I would recommend giving the employed models more
  'speaking' names. I found it hard to keep track of all of them and had
  to refer to the appendix constantly.

- For Figure 4, please show the full space, together will all charts

- p.7: please give some ideas (see above) for how to use the covering of
  the points in practice; I like that the object can be reconstructed
  with a proper set of charts, but the paper could make the necessity
  of the technique much more obvious by choosing stronger examples.

- p.7: the object arguably *also* has a complex geometry, not only
  complex topology. This should be mentioned.

- p.8: the discussion of MNIST is slightly incorrect; as outlined above,
  many digits appear to be generated by multiple charts, while some,
  such as `1` do not appear on more than one chart.

- The metrics in Section 5.3 should be introduced earlier, maybe at the
  expense of some exposition in the introduction or the simpler
  examples; it is not good style to have to refer to the appendix to
  understand a core experiment of a paper.

- p.8: I do not understand the term 'wholly pyramid'.

- p.11: the decoder should map to $x_i$, if I am not mistaken

- p.11: I would suggest a more consistent terminology to describe the
  models. The prediction function is replicated multiple times, for
  example, so why not introduce a shorthand notation for this?

- p.12: '\cup' and '\cap' need to be switched: the *intersection* of
  domains needs to be empty, not their *union*

- p.13: to what extent are the 'faithfulness' and 'coverage' established
  metrics? It seems that they are developed for this paper, so I would
  explain them in the main text and also make clear why they are
  desirable metrics---else, the metrics could be criticised as being
  fine-tuned for the proposed method.

  For example, if *coverage* can measure the phenomenon of *mode
  collapse*, this needs to be demonstrated.

## Minor comments

Some typos:

- low dimensional --> low-dimensional
- eigen-functions --> eigenfunctions
- considers manifold point --> considers a manifold point
- paring subnetwork --> pairing subnetwork
- paramterized --> parametrized
- preformed --> performed [occurs multiple times]
- chats --> charts
- Lipshitz --> Lipschitz (in Figure 4)
- evalutation --> evaluation
- seciton --> section

**Experience Assessment:**

I have published in this field for several years.

**Review Assessment: Checking Correctness Of Derivations And Theory:**

I carefully checked the derivations and theory.

**Review Assessment: Checking Correctness Of Experiments:**

I carefully checked the experiments.

**Review Assessment: Thoroughness In Paper Reading:**

I read the paper thoroughly.

---

> ### Author Response · Authors · 2019-11-15
> **Response to reviewer 1, part 1**
>
> We thank the reviewer for their extremely thorough and thoughtful review.
>
> Concern 1. Response: a). Claim ‘triviality’ of Euclidean space.
>
> When we say that Euclidean space is trivial, we mean that it has the flat metric (‘trivial’ metric here is a mathematical terminology referred as a flat metric. We’ve revised ‘trivial’ as ‘flat’ in our revision) . We agree that many models with Euclidean space do a very good job of reconstructing data which may have very complicated structure. In fact, this is not in conflict with our geometric explanation of encoding and decoding as a flat Euclidean domain can be used to parameterize smaller region of a complicated manifold.  We are most interested in how the structure of the data relates to the structure of  latent space, and how we can recover geometric information (which is uniquely determined from the charts and transition functions) by training a model.
>
> b). Terminology of ‘Topology’ and ‘Geometry’.
>
>  Topology of the data is referred to  global properties (such as holes in the manifold). We intentionally use geometry since the learning results of encoders and decoders provide a parameterization of the data manifold. This can further help us understand geometric information of manifolds including geodesics, curvature etc. Our revision has clarified this terminology.
>
> c).  Initial Encoder is not analyzed.
>
> The initial encoder serves as a dimension reduction step to find a low dimensional isometric embedding of the data. For example, given a data manifold as a torus embedded in 1000D space, the initial encoder reduces the dimension  from 1000D to a lower ambient space (ideally 3D in this case), and the chart encoders map from 3D ambient space to 2D charts as the intrinsic dimension of the torus is 2.  This helps us to save parameters and reduce computational costs. Another example is if a data point was an extremely high-resolution photograph, it would not be unreasonable to down sample it before passing it into a classification network. It is certainly possible that this down sampling will lose some geometric information. Since the chart parameterization can preserve the topology of the low dimensional manifold and loss function encourages exact reconstruction of the high dimensional manifold, it is not unreasonable to expect that the most important features are preserved.
>
>
> Concern 2 Response.  Questions on number of charts.
>
> We agree that there are many possible chart parameterizations with different number of charts which are all equally valid for a given manifold. There is, however, a lower bound on the number of charts needed to cover some manifolds (for example, at least two charts are needed to cover a sphere). Our original scheme to choose the number of charts to use is to overestimate the number needed and then add a regularization which encourages some chart encoders to die off. We have included a new experiment in the appendix which illustrates this scheme.
>
> Concern 3 Response: a). More parameters leads to better results?
>
>  Our aim is manifold inspired models with latent spaces that are lower dimensional than those of traditional encoders and are closer to the intrinsic dimension of the data. The decoupled nature of the decoding operations mean that our models will tend to be larger in terms of number of parameters. We partially agree that more parameters will result in better reconstruction loss. However, the double torus example (used in the introduction and detailed in A10) shows that increasing the number of parameters in a VAE alone (without increasing the latent dimension) does not allow one to simultaneously produce good reconstruction and generation. A latent space which has too small of a dimension will not be able to cover a manifold, and one which is too large will generate points far from the data manifold. Thus the structure of latent space is more important than the number of parameters. This is one of the main objectives of this paper.

---

> ### Author Response · Authors · 2019-11-15
> **Response to reviewer 1, part 2**
>
> b).  Uniformity of Covering
>
>   In general, when we say that the charts need to be balanced and regular, we do no mean uniform. We mean that the distortion of the encoding (and decoding) map should be bounded. Rather than have many charts which approximate the entire manifold poorly, we would like local charts which describe a neighborhood of the manifold very accurately. The size of the charts depends on the geometry of the data. Flat regions can be covered with very large charts, but areas of high curvature may need multiple charts. Therefore uniformly sampling in the latent space does not correspond to uniformly sampling on the data manifold. The observation that some classes appear more than others is not problematic, as the chart selection module does not require equally balanced classes and some of these points may lie in the intersection of several charts. This flexibility is an advantage of our method, since it uses a data-driven approach to determine the coverage of the local neighborhoods on the manifold. The opening examples on the circle, sphere and double torus result in relatively uniform charts because the underlying manifolds are highly regular.
>
> c). Digital morphing example.
>
>  Our purpose with this experiment is not to claim that we are ‘better’ at this type of application, just that we are able to do it without using a variational assumption on the latent space. In the standard VAE set up this assumption prevents the model from memorizing the data-our method does so via the geometrically inspired architecture and Lipschitz regularization.
>
> d) Why use chart?
>
> The main contribution of introducing the chart structured space is to allow chart-structured latent space to reflect data manifold structure. This is necessary according to geometric argument and our numerical experiments. Moreover, the structure of the data manifold will always be compatible with some chart space. A chart representation uniquely identifies the manifold up to isometry. This means that any geometric property that we wish to study---geodesic distance, curvatures etc---can all be formulated in terms of the chart representation. We’ve added an experiment showing how to estimate geodesics on a data manifold from a trained CAE. .
>
> Technical clarity Response:
>
> a). Why Euclidean chart do not suffer from being too simplistic.
>
>  In this work the emphasis is placed on multiple charts (as opposed to one chart), rather than non-Euclidean (as opposed to Euclidean). Models with a single Euclidean latent space can do a very good job of reconstructing data which may have very complicated structure, but these representations do not capture all global properties of the data. Referring to our motivating example there is no way for a standard auto-encoder to detect the cyclic structure of a circle, even though it may be able to reconstruct all points on it. According to differential geometry, any compact manifold can be covered with a finite collection of charts, each of which is homeomorphic a euclidean domain. This motivates us to use a collection of simple euclidean domains as building blocks to represent complicated structures.
>
> b) about homeomorphism.
>
>  We use the mathematical concept of the atlas parameterization as a motivation for the architecture and loss functions.We view the proposed chart auto-encoder as being approximately homeomorphic in the same way that auto-encoders are viewed as being approximately invertible. Moreover, the purpose of using a chart representation is that a homeomorphism always exists between a local neighborhood on the manifold and a Euclidean domain. In addition a compact manifold can always be covered with a finite collection of these charts which obey the chart transition conditions.
>
> c) The torus embedding example.
>
>  The double torus cannot be isometrically embedded in R^2, so no matter what representation is learned, some geometric information (distances, curvatures, etc) will be lost. Furthermore topological structures cannot be preserved either because the torus is not homeomorphic to R^2.
>
> d)- p.2: paths become invariant to _what_ exactly?
>
>   In a VAE whose latent dimension is larger than the intrinsic dimension of the manifold, at any point in the latent space, there are directions in which moving either: does not change the output of the decoder or, moves off the manifold.
>
> e). -p.2: what is the 'topological class'?
>
> Here we just mean the set of manifolds with the same topology.
>
> f) p.2: would LLE not be a good precursor?
>
> Similar to ISOMAP, LLE may help motivate this paper. Both of these techniques use a single operation to encode the entire manifold, while our approach automatically learns multiple local operations.

---

> ### Author Response · Authors · 2019-11-15
> **Response to reviewer 1, part 3**
>
> g) is this paper to be seen as an implementation of Chen et al. (2019)?
>
>  No, This is not an implementation of Chen et al 2019. Their paper deals with using a network to represent a function on a known, fixed manifold, while we are concerned with capturing the manifold.
>
> h) the concept of intrinsic dimension slightly varies in literature.
>
> We have clarified this definition in our revision.
>
> i) Section 4.1 mixes 'topological' and 'geometrical' concepts.
>
> In this case we do mean geometry, sorry for our typo.
>
> j) p.5: I like the 'partition of unity' approach…
>
> This  looks like a convex combination, but it is more restricted than a convex combination. Each of p_a has compact support in U_a, i.e. p_a(x) has contribution as long as x in the compact support of p_a.
>
>  k). - p.13: to what extent are the 'faithfulness' and 'coverage' established metrics?
>
>  The reconstruction error measures the fidelity of the model, the unfaithfulness measures how far synthesized data decoded from samples drawn on the latent space are to samples from the original training data and coverage indicates how much of the training data is covered by the encoder. Models which produce unrealistic data when sampling from the latent space will have high unfaithfulness sores and models which experience mode collapse will have low coverage scores. Metrics similar to the faithfulness have been previously used to measure novelty, but coverage is established metric for measuring mode collapse in GANs.

---

### Official Review · AnonReviewer3 · 2019-10-22
**Official Blind Review #3**

**Rating:** 3

**Review:**

Summary:
The authors provide a model that is based on multiple Auto-Encoders, and it is claimed that each Auto-Encoder learns a local chart map of the data manifold.

In general the paper is ok written and tries to present the idea and the theoretical background in a nice way. However, there are few things that in my opinion should be improved (see comments). More importantly, at the first sight the proposed model seems to be solid, but I think that there are some details, which make the model to not behave as it should in theory.

Comments:

1. As regards the related work, I think that some references should be included. For instance, several recent papers which discuss the topological structure of the latent space [1,2,3,etc]. Also, recently the geometry of the latent space is analyzed through Riemannian geometry and points out that the uncertainty of the generator is crucial for capturing correctly the data manifold structure [4]. Moreover, there is some work where multiple generators are used in order to model the data [5].

2. I am not entirely convinced that the proposed model learns the charts of the manifold. Instead, I think it just utilizes several auto-encoders, and each of them specializes in some parts of the data manifold:

- First of all, the chart map is very well defined operator. A point in the intersection of two neighborhoods on the manifold U_a, U_b that overlap, has to be reconstructed "exactly" by the two corresponding charts. However, from the modeling steps and the experiments I cannot see why this is the case.

- As regards the technical details. The loss function of Eq. 2 essentially implies that only one chart is specialized for the sample x. However, such that to have chart maps, there should be samples on the intersections of neighborhoods on the manifold that are reconstructed by both charts. I do not see how the proposed model can tackle this issue.

- The loss function of Eq. 3 is even more debatable. The reason is that a mixture of auto-encoders is used to reconstruct the point x. However, this is not the definition of a chart. This is simply a way to use several auto-encoders to reconstruct the data, where the function p_i(x) (acts as soft assignment) chooses which of the auto-encoders should be used for the sample.

- The chart is defined as an invertible map. I can understand that in practice the decoder is considered as the inverse of the chart. However, we cannot guarantee that there are not cases where the decoder creates a surface with intersections or that "degenerates" some parts of the surface (instead of a 2-dimensional surface, it generates an 1-dimensional curve). In this case, the chart is not invertible on these parts of the surface. So I am not sure if we can directly consider an auto-encoder based on Neural Network as chart map.

3) I think that the first global encoder E couples all the other encoders. Also, it is stated by the authors that this step should respect the manifold topology, which in general is not the case. So even if this helps for computational efficiency, it does not respect the theory.

4) The pretraining together with the regularization make me to believe that the model first separates the dataset into K clusters, and then learns an approximately linear model for each cluster. I think that this is what the Eq. 2 implies, or a soft assignment (weighted) version if the Eq. 3 is used.

5) In the experiments some of the previously mentioned issues appear:

- In Fig. 5 seems that few of the charts are more "important" than the others. However, I am more curious for what happens on the intersection of the neighborhoods on the manifold. For instance, from the figure it seems that some of the U_a on the surface are "disconnected". Does this mean that simply some of them (e.g. U_a and U_b) intersect and thats why the disconnected sets (e.g. of U_a) appear? If this is the case, then how the two chart maps behave on the intersection?

- As regards the MNIST, I do not think that it is a good example to support the chart learning. Most probably, there are 10 (disconnected) manifolds, and each of them should be modeled by a particular chart (or many charts per digit-manifold). In this case I think that the p(x) should be exactly 0 and 1, such that to chose one chart per digit. Also, in the current setting of the experiment, essentially all the data are considered to lie on the same data manifold. So what is the behaviour of the charts in the parts of the ambient space where there are no data?

- I think that is very important a well constructed experiment that shows the behaviour of the charts on overlapping domains (Sec A.3). Even an example in 2D ambient space with embedded 1-dimensional (disconnected) manifolds.

- Why in Fig. 7 some bars are missing? Also, from the appendix it seems that the CAE ||| is a very powerful model with 10 latent spaces of 25 dimensions each, and moreover, is the only one that uses convolutions. Since, from the text is not clear if the VAE II uses convolutions, and also, it has only one 25 dimensional latent space. In my opinion this is not a reasonable comparisson.

- Probably comparisons with other models that use multiple generators or even latent spaces that respect the topology of the data manifold could be included.

Minor comments:
1) In my opinion, from the first paragraph of Sec 4.1. is not clear how the function p(x) is defined.

2) The regularization part is a bit unclear. Strong regularization on the decoders probably means that locally the auto-encoders will behave approximatelly as linear models? Especially, since the initialization is based on local PCAs (pre-training), which can potentially act as an inductive bias. Also, in the beggining of paragraph 3 in Sec 4.2. it is stated that the Lipschitz regulariation is used for the decoders, but next it is introduced for the encoders. This needs clarification.

3) How is defined the term at the end of Eq. 5?


In general, I like the problem that the paper aims to solve. However, I have the feeling that the proposed approach is quite debateable. Instead of chart learning, in my opinion, I think that the model just uses several auto-encoders and each of them is specialized at different subsets of the training data. These subsets are chosen at the pre-training phase, and then the function p(.) acts as an (soft) assignment function. Overall, my main question is what happens on the overlap of two neighborhoods on the data manifold? Also, what happens if the data lie on disconnected components?

References:
[1] Diffusion Variational Autoencoders, Luis A. Perez Rey, et al., 2019.
[2] Hyperspherical Variational Auto-Encoders, Davidson, Tim R., et al., 2018.
[3] Hierarchical Representations with
Poincaré Variational Auto-Encoders, Emile Mathieu, et al., 2019.
[4] Latent Space Oddity: on the Curvature of Deep Generative Models, Georgios Arvanitidis, et al., 2018.
[5] Competitive Training of Mixtures of Independent Deep Generative Models, Francesco Locatello, 2019.


**Experience Assessment:**

I have published one or two papers in this area.

**Review Assessment: Checking Correctness Of Derivations And Theory:**

I assessed the sensibility of the derivations and theory.

**Review Assessment: Checking Correctness Of Experiments:**

I assessed the sensibility of the experiments.

**Review Assessment: Thoroughness In Paper Reading:**

I read the paper at least twice and used my best judgement in assessing the paper.

---

> ### Author Response · Authors · 2019-11-15
> **Response to Reviewer 3, Part 1**
>
> 1. As regards the related work.
>
> R3.1. Response: Thank you for the references, we have included these in the updated version of the paper.
>
> 2. I am not entirely convinced that the proposed model learns the charts of the manifold. Instead, I think it just utilizes several auto-encoders, and each of them specializes in some parts of the data manifold….
>
> R3.2. Response: If several auto-encoders cover a different parts of a manifold and obey the chart transition conditions, then they are essentially forming an approximation of an atlas. However, the challenge is in dividing the data and learning the transitions.  Our methodology is to use neural networks as universal approximators and then use the architecture, loss functions and regularization to mimic charts. For example, a point x which is in the overlap of two charts U_a and U_b will have both the terms ||x-x_a|| and ||x-x_b|| in the loss. Then we will have x_a = x_b when the model is minimized.
>
> 3. The loss function of Eq. 2 essentially implies that only one chart is specialized for the sample x…..
>
> R3.3. Response: In eq (2), the gradient from the first term is only passed to one of the decoders, but the second term ensures that other decoders in the overlap are also updated. For example, if a point x is in the intersection of U_a and U_b then p_a and p_b will both be larger than the rest of the p_i. Then the weights on l_a and l_b will be largest and ||x-x_a|| and ||x-x_b|| will dominate the second term in the loss. We have written this loss function more clearly.
>
> 4. The loss function of Eq. 3 is even more debatable….
>
> R3.4. Response: We agree that this acts essentially as a soft assignment or a convex combination of each of the decoders. This idea is based on the partition of unity from topology and can be used instead of the chart prediction model using the same architecture. Here each of the p_i’s are only non-zero in the U_i region, this is illustrated in the new experiment in the appendix A7. This is different from ensemble networks which separates the image into different parts to be encoded by different encoders.
>
> 5. Varying local dimension.
>
> R3.5. Response: In general working with manifolds of varying local dimension (or multiple manifolds of different dimensions) is a very hard problem which we do no look to solve in this paper. The problem of these ‘degenerate points’ are equally changing for standard models. We use manifold concepts as theoretical motivation to study the geometric structure of the latent space. Our numerical experiments validate the effectiveness of this interpretation.
>
> 6.  I think that the first global encoder E couples all the other encoders. Also, it is stated by the authors that this step should respect the manifold topology, which in general is not the case. So even if this helps for computational efficiency, it does not respect the theory.
>
> R3.6. Response:  The initial encoder serves as a dimension reduction step to find a low dimensional isometric embedding of the data. For example, given a data manifold as a torus embedded in 1000D space, the initial encoder reduces the dimension  from 1000D to a lower ambient space (ideally 3D in this case), and the chart encoders map from 3D ambient space to 2D charts as the intrinsic dimension of the torus is 2.  In practice we observe that global encoder can save parameters and reduce computation costs for high dimensional data. Another example is if a data point was an extremely high-resolution photograph, it would not be unreasonable to down sample it before passing it into a classification network. It is certainly possible that this down sampling will lose some geometric information. Since the chart parameterization can preserve the topology of the low dimensional manifold and loss function encourages exact reconstruction of the high dimensional manifold, it is not unreasonable to expect that the most important features are preserved.
>
> 7.  The pretraining together with the regularization make me to believe that the model first separates the dataset into K clusters, and then learns an approximately linear model for each cluster…
>
> R3.7. Response: We respectfully disagree.  The pretraining works to ensure that each of the decoders is ‘on’ the manifold so that when training begins there are no decoders which are always inactive. Since the chart selection module is learned in conjunction with the rest of the model there is no prior segmentation of the data. During training the charts will move, change sizes, overlap or disappear. This is quite different from clustering  the data first and training encoders independently on each cluster. Additionally, our numerical experiments show that the models are not nearly linear (see fig 1,4,5).

---

> ### Author Response · Authors · 2019-11-15
> **Response to Reviewer 3, Part 2**
>
> 8.  In Fig. 5 seems that few of the charts are more "important" than the others….
>
> R3.8. Response: Because this example has non-constant curvature we do not expect that the U_i’s are of uniform sizes. Even though each chart is simply connected, their intersections may be disconnected. However, since the charts should agree in this transition zone this is not an issue.  We have included a new set of visualization in the appendix which illustrate the transition zones in the appendix A7.
>
> 9. As regards the MNIST, I do not think that it is a good example to support the chart learning….
>
> R3.9 Response: We respectfully disagree.  If the data were indeed 10 disconnected manifolds then p(x) should behave as an indicator.  However, it is very likely that there is a continuous path between “1”s and “7s” or between ‘6’s and ‘8’s. Since we do not assume any ground truth labels the network must learn how to handle these transitions. If the network is properly trained then the each of the charts will always map onto the manifold. However, if a point from off the manifold is passed into the encoder after the model has been trained, the output will be on the manifold rather than being close to the original input.
>
> 10. I think that is very important a well constructed experiment that shows the behaviour of the charts on overlapping domains ...
>
> R3.10 Response: Thank you for this comment. We have added a test showing the behavior in a transition zone in the appendix A7..
>
> 11. Why in Fig. 7 some bars are missing?
>
> R3.11 The CAE III model begin with convolution layers, but VAE II does not, this is explained in the appendix A.2 and A.7.  We have made the distinction clearer in the revision. For the sphere data, it is not clear how to use a convolution. The convolution works to extract features from a single training example, which would just be a single point in R^3 for the sphere. Analogously, a point from the MNIST manifold is an image which can be convolved on.
> The revision updates results for our simplest models on SVHN.
>
> 12.  In my opinion, from the first paragraph of Sec 4.1. is not clear how the function p(x) is defined
>
> R3.12: The exact definition of p(x) depends on the exact architecture and loss functions used. We've clarified this part in the revision.
>
> 13. The regularization part is a bit unclear…
>
> R3.13: The Lipchitz regularization acts as a bound of the spectral norm of the decoders. This promotes smoothness of each decoder and allows them to balance out (rather than having the entire manifold dominated by a single chart). We view this initialization as incorporating our domain knowledge which can help us improve the training stability. Bounded Lipschitz models do not mean they are approximately linear. We use the regularization for the encoders and decoders as they are models as inverses of each other.
>
> 14. How is defined the term at the end of Eq. 5?
>
> R3.14: delta_ab is the Dirac delta, we’ve clarified this in the revision.

---

### Author Response · Authors · 2019-11-15
**Executive Summary of Revision**

We’d like to thank all reviewers for their thoughtful remarks and contributions. We have uploaded a revision which addresses comments raised in the review process.

There are several main changes which we have made in the revision. Major changes have been marked in magenta, while typos and grammatical errors have been corrected inline.

Firstly, we have re-written section 4.1 to better describe the properties of our chart prediction module and how we deal with points which fall into the overlapping regions. We have also added an experiment in the appendix to illustrate the behaviour of the network in these transition zones.

Next, we have expanded the discussion on our network regularization and provided an additional example (in the appendix) of our method to choose an appropriate number of charts.

In the experiments section, we have added the requested test for simple models on the MNIST and SVN test sets. Additionally, we have expanded the discussion of the metrics we use to validate our tests. The ‘faithfulness’ metric has also been re-named ‘unfaithfulness’ to reflect that a low score is desirable.

Last but not least, we have added several experiments in the appendix to answer reviwers’ comments and clarify the proposed method. In A.7, we included an illustrative example to show the overlapped regions and transition functions. In A.8, we demonstrated the automatic removal of over-estimated charts. In A.9, we used a synthetic example on approximating geodesic on a data manifold based on our learning results. This indicates a great potential of the charts structure, i.e. it helps us understand geometry information of data manifold. In A.10, we added an experiment to show a single chart VAE can not have good generating results even with more complex network (with more parameters).

---

### Decision · Program_Chairs · 2019-12-19

**Decision:**

Reject

**Comment:**

This paper proposes to use more varied geometric structures of latent spaces to capture the manifold structure of the data, and provide experiments with synthetic and real data that show some promise in terms of approximating manifolds.
While reviewers appreciate the motivation behind the paper and see that angle as potentially resulting in a strong paper in the future, they have concerns that the method is too complicated and that the experimental results are not fully convincing that the proposed method is useful, with also not enough ablation studies. Authors provided some additional results and clarified explanations in their revisions, but reviewers still believe there is more work required to deliver a submission warranting acceptance in terms of justifying the complicated architecture experimentally.
Therefore, we do not recommend acceptance.